# Electrolyzed–Reduced Water: Review I. Molecular Hydrogen Is the Exclusive Agent Responsible for the Therapeutic Effects

**DOI:** 10.3390/ijms232314750

**Published:** 2022-11-25

**Authors:** Tyler W. LeBaron, Randy Sharpe, Kinji Ohno

**Affiliations:** 1Centre of Experimental Medicine, Institute for Heart Research, Slovak Academy of Sciences, 841 04 Bratislava, Slovakia; 2Molecular Hydrogen Institute, Enoch, UT 84721, USA; 3Department of Kinesiology and Outdoor Recreation, Southern Utah University, Cedar City, UT 84720, USA; 4H2 Analytics, Henderson, NV 89052, USA; 5Division of Neurogenetics, Center for Neurological Diseases and Cancer, Nagoya University Graduate School of Medicine, Nagoya 466-855, Japan

**Keywords:** electrolyzed–reduced water, ionized water, alkaline water, alkaline-ionized water, hydrogen water, functional water, alkali ion water, alkaline-reduced water

## Abstract

Numerous benefits have been attributed to alkaline-electrolyzed–reduced water (ERW). Sometimes these claims are associated with easily debunked concepts. The observed benefits have been conjectured to be due to the intrinsic properties of ERW (e.g., negative oxidation–reduction potential (ORP), alkaline pH, H_2_ gas), as well enigmatic characteristics (e.g., altered water structure, microclusters, free electrons, active hydrogen, mineral hydrides). The associated pseudoscientific marketing has contributed to the reluctance of mainstream science to accept ERW as having biological effects. Finally, through many in vitro and in vivo studies, each one of these propositions was examined and refuted one-by-one until it was conclusively demonstrated that H_2_ was the exclusive agent responsible for both the negative ORP and the observed therapeutic effects of ERW. This article briefly apprised the history of ERW and comprehensively reviewed the sequential research demonstrating the importance of H_2_. We illustrated that the effects of ERW could be readily explained by the known biological effects of H_2_ and by utilizing conventional chemistry without requiring any metaphysical conjecture (e.g., microclustering, free electrons, etc.) or reliance on implausible notions (e.g., alkaline water neutralizes acidic waste). The H_2_ concentration of ERW should be measured to ensure it is comparable to those used in clinical studies.

## 1. Introduction

Electrolyzed–reduced water (ERW), also called “alkaline ionized water”, is a type of alkaline water produced from water electrolysis. The devices that perform this are known as “alkaline water ionizers” or “water electrolyzers” [1]. ERW has been a popular type of water for many decades and continues to be popularized across the world due to its many claimed health benefits. Some of the claims have a scientific rationale and are supported by published studies, whereas others lack evidence or even falsifiability [1]. Over the past 50+ years of ERW use and research, it has now been confirmed that ERW can exert meaningful biological effects and that molecular hydrogen is responsible for those biological effects [2]. Unfortunately, the recognition of the importance of H_2_ in ERW is still lagging. This has contributed to several unfortunate consequences including (i) the difficulty for mainstream science to accept that ERW has biological effects, (ii) the persistence and proliferation of debunked and pseudoscientific claims about ERW, (iii) inadequate scientific research on ERW, (iv) health-conscious consumers being taken advantage of when purchasing suboptimal devices, and (v) ignoring several safety concerns or even promoting them as benefits.

The electrolysis of water is a common method of producing hydrogen gas [3]. This makes water ionizers a potentially effective method for making hydrogen water. However, a negative ORP, an alkaline pH, and a reduced oxygen gas concentration, which inevitably occur in making ERW with electrolysis, blur the identity of the effective agent in ERW. This article provides a historical overview of ERW and the development of the most prominent claims, which are then juxtaposed with the scientific consensus. We first discuss some of the properties and biological effects of molecular hydrogen, followed by a review of the evidence that illustrates that molecular hydrogen is indeed the exclusive beneficial agent in ERW.

## 2. Physicochemical Properties and Biological Effects of Molecular Hydrogen

Molecular hydrogen is a diatomic gas consisting of two hydrogen atoms covalently bonded together. It is a stable neutral molecule that is explosive between a 10% and 75% concentration [4], but not when dissolved in water. The cellular bioavailability of molecular hydrogen is extremely high due to its unique physicochemical properties. Its small size, low mass, neutral charge, and nonpolar nature, coupled with its high rate of diffusion, allow it to easily penetrate cellular biomembranes and diffuse into the mitochondria and nucleus [5]. Indeed, molecular hydrogen is the smallest molecule, with a covalent radius of 31 pm, which is less than half the size of oxygen gas. It is the lightest molecule (2.0159 g/mole) and the least dense. Hydrogen can be dissolved in water up to 0.8 mM (1.6 mg/L) at standard ambient temperature and pressure (SATP). However, it is several times more soluble in lipids [6]. When dissolved in water, it does not dissociate into electrons and protons regardless of the pH but is simply surrounded by water molecules, forming aqueous H_2_. Hydrogen gas comprises about 5.50 × 10^−5^ percent of atmospheric pressure; therefore, at equilibrium, the dissolved H_2_ concentration in ambient water is expected to be 4.29 × 10^−7^ mM (8.65 × 10^−7^ mg/L). This is less than one millionth of a milligram per liter; in contrast, saturated hydrogen water contains 1.6 mg/L. A concentration of 1.6 mg/L may not seem significant, but because H_2_ is the lightest and smallest molecule, it should be compared using moles rather than mass. The ingestion of 1 L of H_2_-saturated water provides more “therapeutic moles” than would the ingestion of a 100-mg dose of vitamin C (0.79 millimoles H_2_ vs. 0.57 millimoles vitamin C) [7]. Similarly, a normal 3-mg dose of melatonin, which is only around 3% bioavailable [8], results in only 0.09 mg reaching the blood (or 21 millimoles), which is 26 times less than the moles of molecular hydrogen following the ingestion of one liter of saturated H_2_ water.

Hydrogen gas has recently been extensively studied for its unique antioxidant, anti-inflammatory, and anti-cellular-stress effects. Clinical studies on hydrogen have demonstrated its potential application for cognitive impairment [9], metabolic syndrome [10], stroke [11], COVID-19 [12], and in exercise medicine [7]. The clinical and molecular effects of hydrogen gas have been extensively reviewed [13,14].

## 3. Water Ionizer History and Its Connection with Hydrogen

Water electrolysis is the electrical decomposition of water into hydrogen and oxygen gas [i.e., electricity + 2H_2_O→2H_2_(g) + O_2_(g)] [15]. This electrochemical reaction is an oxidation–reduction reaction where hydrogen gas is formed at the cathode by the reduction of hydrogen ions [2H^+^(*aq*) + 2e^−^→H_2_(*g*)], and oxygen gas is formed at the anode by the oxidation of hydroxide ions [2OH^−^(*aq*)→O_2_(*g*) + 2H^+^(*aq*) + 2e^−^] [15]. By separating the two electrodes with a semi-permeable membrane that is impervious to H^+^ and OH^−^ ions, one can make alkaline water containing hydrogen gas at the cathode, and acidic water containing oxygen gas at the anode [1]. The water produced at the cathode is referred to as “electrolyzed reduced water” (ERW). “Electrolyzed” is used because it went through electrolysis and “reduced” is used because of the reduction reaction at the cathode.

ERW is also often referred to as “alkaline ionized water”. The term “ionized” is arguably a misnomer because the water molecules themselves are not ions (i.e., only H_2_O and not H_2_O^+^ or H_2_O^−^); the water goes through a process of ionization to form more H^+^ ions at the anode and more OH^−^ ions at the cathode. Moreover, although the produced water contains more or less H^+^ and/or OH^−^ ions, making the neutral-pH source water alkaline or acidic, the product of the OH^−^ and H^+^ concentration is still equal to the ionic product of water (i.e., K_w_ = [OH^−^][H^+^] = 1 × 10^−14^). Thus, the produced water does not contain any more ions than the original water, conforming to the law of electroneutrality, and the water itself is not really ionized.

ERW is characterized by having a high pH, low dissolved oxygen gas concentration, high dissolved hydrogen gas concentration, and a negative oxidation–reduction potential (ORP) [1,16,17,18,19,20,21]. In 1931, research on ERW in Japan began [22], and in 1965 the Japanese government approved water ionizers to help alleviate various gastrointestinal conditions [1,23,24,25]. A decade later, in 1978, the Korean Federal Drug Administration also approved water ionizers for similar reasons [26]. However, in these early years, the role of solubilized hydrogen gas as a therapeutic agent was not yet recognized. This is understandable since this was before it was scientifically determined that molecular hydrogen had any biological effects.

### Observed Benefit from ERW

Since at least May 1985, there have been many documented clinical improvements from the consumption of ERW, which were thought to be miraculous at the time [27]. Around 1997, ERW research increased, and additional studies revealed that ERW has antioxidant effects [28], protects DNA from oxidative damage [17,29,30,31], promotes GLUT4 translocation [32], protects pancreatic beta cells from alloxan-induced cell damage [33], prevents aspirin-induced gastric mucosal injury [34], suppresses lipid peroxidation [35], and has anti-diabetic [36,37,38,39] and anticancer [40,41,42,43,44] effects.

Nevertheless, the primary therapeutic agent in ERW was still unknown [28,31,33]. Molecular hydrogen was viewed as simply an inert byproduct of electrolysis with no biological value [45]. The effectiveness of ERW was estimated by pH and ORP but not by the concentration of dissolved hydrogen gas. Indeed, most of these early studies did not report the concentration of the dissolved hydrogen gas until post-2007, which was when the therapeutic effects of H_2_ were first clearly demonstrated [46].

## 4. Rejected Hypotheses on ERW

Since hydrogen gas was only seen as a biologically inert byproduct of electrolysis, many conjectures were made as to what property in ERW was responsible for its therapeutic effect. For example, (i) alkaline pH, (ii) increased energy of the water reflected by a claimed higher K_w_ of ERW compared to tap water [28,45], (iii) increased cellular bioavailability of the water due to a claimed different water structure (e.g., microclustering) [26,47,48,49,50,51,52], (iv) negative ORP, which was viewed as available electrons for quenching radicals [50,53,54,55], (v) platinum nanoparticles, and (vi) atomic hydrogen (referred to as active hydrogen), which was proposed because molecular hydrogen was viewed as too inert [17,56]. Table 1 summarizes the frequently made claims regarding ERW and provides a brief comment on their scientific plausibility. The table focuses only on the physico-chemical properties of ERW and not on the many biological health claims, which are far too numerous and beyond the scope of this article. Moreover, many of the health claims become irrelevant once either the physical/chemical properties of ERW are scrutinized/refuted (e.g., the negative charge of water improves cellular voltage, the altered water structure improves cellular hydration, etc.) and/or it is demonstrated that molecular hydrogen alone can explain the observed therapeutic biological benefits.

The unfortunate proliferation of these claims has also led to ERW being referred to as: ion water, ionic water, negative water, electron water, micro water, microclustered water, cluster water, nano water, hexagonal water, hydroxyl water, miracle water, energized water, electrically vibrational water, etc. Importantly, as will be discussed, all the “hypotheses” listed in Table 1, except for the more recent H_2_ explanation, fail to hold up under scientific scrutiny. The next section will discuss a few of the most prominent conjectured propositions.

### 4.1. Alkaline pH

Normal healthy plasma pH is ≈7.4 (i.e., 7.35 to 7.45), which is slightly alkaline. If the pH deviates out of this narrow range, pathological consequences can arise. A cellular pH outside the range of ≈6.8 to 7.8 is incompatible with human life because otherwise the confirmation/structure of critical proteins change, resulting in their loss of function. The body continuously fights to maintain this tight pH range, which primarily is a fight against acid (i.e., H^+^ ions) produced during the metabolism of food. Many of these hydrogen ions eventually combine with oxygen in the mitochondria to form water (e.g., 4H^+^ + O_2_→2H_2_O). However, not all of them do, which increases the H^+^ ion concentration (lowers the pH). Additionally, during the process of metabolism, CO_2_ is produced, which further lowers the pH of the blood by adding more H^+^ ions according to the reaction (CO_2_ + H_2_O→HCO_3_^−^ + H^+^). The amount of H^+^ produced per day simply by the production of CO_2_ in normal metabolism (i.e., oxidation of fats, carbohydrates, and proteins) would create an H^+^ concentration over one hundred million times as concentrated as the normal plasma H^+^ concentration.

It is well known that certain diseases that increase acid production (e.g., diabetic ketoacidosis and lactic acidosis due to ischemia), loose bicarbonate (e.g., chronic diarrhea and Addison’s disease), or reduce renal acid excretion (e.g., renal failure and renal tubular acidosis) can result in a medical crisis due to a lowering of the blood pH. It can be stated that if the blood pH is lower than normal, then the person has a disease. It can also be stated that a lower pH can cause many unfavorable symptoms including headache, joint pain, muscle weakness, heart issues, and impaired cognitive function. However, these correlational facts are often misinterpreted as a unidirectional causality to mean that: (1) an acidic pH “causes” all diseases, and (2) anyone experiencing any of the aforementioned symptoms does so because they have an acidic pH.

Accordingly, it is largely these illogical beliefs that lead to the false conclusion that preventing the acidic pH from occurring also prevents and even cures any disease as well. Therefore, the false conjecture continues, i.e., that drinking ERW that has a high alkaline pH will directly contribute to preventing and curing disease by maintaining/increasing the blood pH to healthy levels.

There are three major fallacies in this line of reasoning:
*False Claim 1: An Acidic pH Causes All Diseases*

This statement comes from the recognition that those with some diseases (e.g., diabetes) may have a low pH, and a low pH can cause other diseases. However, it should be recognized that it was the disease (i.e., diabetes) that caused the low pH, not the low pH that caused the disease. Indeed, many diseases do not influence the pH of the body at all, while others can induce lethal alkalosis (too high of a pH) such as certain types of cancers, kidney dysfunctions, dehydration, genetic mutations of chloride channels, hyperaldosteronism, as well as Bartter’s, Gitelman’s, and Liddle’s syndromes, etc.
*False Claim 2: Ordinary Sensations as Symptomatic Evidence of a Low pH*

Many of the symptoms of a low pH such as headache, fatigue, stupor of thought, joint pain, etc., are also symptoms of many other diseases, including just normal life. In fact, many of these same symptoms of too low of a pH can occur under the conditions of metabolic alkalosis. Unfortunately, many people marketing their alkaline products use these symptoms and others to essentially diagnose the consumer with having “acidosis”, and then offer the “cure”, which is the alkaline product they are selling.
*False Claim 3: Drinking Alkaline Water Influences Blood pH and Provides Health Benefits*

It is true that, for some medical conditions, the ingestion of alkaline components (e.g., sodium bicarbonate) may be helpful, for example, heartburn, metabolic acidosis, certain types of kidney stones, etc. However, generally, the ingestion of alkaline components will not provide any medical benefit and can be harmful. Indeed, chronic ingestion of sodium bicarbonate has resulted in life-threatening complications including severe metabolic alkalosis, convulsions, electrolyte imbalance, and rhabdomyolysis [68].

However, even if we accept the suppositive premise that we need to ingest alkaline components to help the body maintain/increase blood pH, this cannot be done with alkaline water produced from ERW. This is because, despite having a high alkaline pH (e.g., ≈9.5), this type of alkaline water has very little buffering capacity (i.e., the ability to resist changes in pH). Indeed, an alkaline pH and buffering capacity are independent properties. For an illustrative comparison, sodium bicarbonate with a pH of ≈8.1 is an effective buffer, meaning it can resist changes in pH. Stoichiometrically, 1 tsp of baking soda (4.8 g) can neutralize as much acid (H^+^ ions) as ≈1800 L of pH 9.5 alkaline water.

Those selling alkaline water machines recognize the low buffering capacity of high-pH alkaline water by demonstrating that a few drops of common carbonated beverages will lower the pH of their alkaline water. However, instead of recognizing that alkaline water is an ineffective “alkalizing” agent, they shift the focus to how “acidic” (and therefore “bad”) the beverage is for the body. These water ionizers do not add alkaline buffering agents to the water, which means that, despite having an “alkaline pH”, the actual “alkalinity” (capacity to resist acidification) is only marginally greater than the original tap water it came from. Accordingly, the alkaline property in ERW will not meaningfully influence the pH of the body any more than the original source water and, therefore, will neither provide beneficial effects nor, most likely, any harmful effects.

On the other hand, it is plausible that naturally occurring alkaline water has meaningful health benefits. However, this is not due to the alkaline pH, but rather to the increased mineral concentration. Specifically, naturally alkaline water typically has higher levels of calcium, magnesium, and other trace elements that are essential for health, and these are highly bioavailable when dissolved in water [69]. Indeed, the results from a meta-analysis show that the ingestion of water containing higher minerals lowers the risks of cardiovascular and other diseases [70].

#### Irrelevance of Alkaline pH in ERW

Importantly, although ERW has an alkaline pH [1,71,72], we can logically exclude this property from being a contributing factor to its therapeutic effects because, in addition to the facts mentioned above, in all the in vitro cell culture studies, the alkaline pH is always neutralized [29,33,35,40,43,44,73,74,75,76,77,78,79,80,81,82]. Otherwise, the high alkaline pH would have injured the cells, since a high pH is incompatible with our cellular structure and function. Despite this fact, many advocates of ERW have incorrectly ascribed the observed benefits of ERW to the alkaline pH. The irony is that the very ERW publications used by marketers as evidence for alkaline pH either do not make any such claims and/or the study actually refutes those claims, since the alkaline pH was neutralized.

### 4.2. Microclustering

Studies on ERW have demonstrated meaningful therapeutic effects, but the explanation and agent responsible for these benefits were previously unknown. Since the alkaline pH property did not make any sense, researchers began testing other properties of the water. One interesting discovery was that when ERW was subjected to ^17^O nuclear magnetic resonance (NMR) testing, the results showed that the bandwidth of ERW was significantly less than tap water. This difference was conjectured to indicate a type of restructuring of the water, specifically the existence of “microclusters” [48,49,52,83]. The claim is often stated that normal water consists of clusters of fifteen or more water molecules, which cannot easily penetrate the cells. ERW, on the other hand, is claimed to be “electrically restructured” to only have three to five water molecules per cluster, making it more hydrating [52,63]. However, the NMR bandwidth change has been shown to be a function of pH, in which any deviation from pH 7 results in a smaller resonance bandwidth [64].

Another commonly used demonstration that attempts to prove the existence of microclusters is the appearance of being able to make tea with ERW. The claim is that smaller water microclusters can more easily penetrate the tea bag and extract the tea molecules, circumventing the normal requirement to heat the water. However, this demonstration only works with some teas, where specifically green tea is the most common tea used. The reality is that the organic molecules in tea, like cabbage and many other foods, serve as natural pH indicators. When the pH is increased, the color turns green and, as the pH is decreased, the solution returns to being clear. This explains why not all teas can be used for this demonstration, as not all teas work as pH indicators. Although it is possible that the higher pH ionizes various tea molecules and increases their solubility and subsequent extraction [84] and that gases (bubbles) may also contribute to phytochemical extraction [85], the main phenomenon is due to the color change from the pH. A similar pH-dependent demonstration relies on mixing oil with “strong alkaline water”. The claim is that the microclustering is responsible for the ability of this high-pH water to mix with oil, when in reality it is simply due to the very high pH, in which the OH^−^ ions react with the ester moieties of the fat to saponify and emulsify the oil. Indeed, this is the same method used for making soap, in which a high-pH substance (e.g., an alkali) is mixed with animal fat. This is the same mechanism behind many “degreasing” products, which also have a high alkaline pH. Another frequent anecdote about drinking alkaline ionized water is that it “feels” that it is absorbed into the body significantly faster. There is no bloating sensation or “sloshing” of the water in the stomach. The mechanism for this apparent phenomenon is often attributed to microclustering. However, if this faster absorption is true, then it would again most likely be attributed to the high alkaline pH of the water. Indeed, gastric emptying is delayed by acidic beverages and increases with a higher gastric pH [86,87].

Indeed, the concept of this type of microclustering in water is refuted by basic principles of chemistry and biology, as well as by specific studies that have tested the microclustering claims [47,51,65]. For example, (i) water molecules do not exist in clusters, and any transient association is fleeting on the femto-time scale (billionth of a second) with no memory of where they were before [88], (ii) water molecules enter the cells in a linear and separated, “not clustered”, fashion through aquaporin protein channels, and (iii) the changes seen in the NMR bandwidth, making of tea, and mixing with oil are all a function of pH, not water cluster size [64].

### 4.3. Oxidation–Reduction Potential

Since at this point ERW researchers had not attributed any of the therapeutic benefits of ERW to its alkaline pH, and there had been no credible research supporting the existence of microclusters, the evaluation of other properties of ERW was required. Like pH, the oxidation–reduction potential (ORP) is an intrinsic property of a solution. The ORP can be measured with an ORP probe comprising a platinum electrode as the working electrode and a reference electrode (usually Ag/AgCl), which is calibrated to the standard hydrogen electrode [59]. The two electrodes measure the potentiometric difference (potential) giving units of volts. Accordingly, solutions that contain highly oxidizing chemical species (e.g., chlorinated water, hydrogen peroxide, chromium trioxide, etc.) give a high positive ORP reading (e.g., +500 to >+1000 mV), whereas solutions that contain reductive chemical species (e.g., 2-mercaptoethanol, hydrogen gas, ascorbate, and other certain antioxidants, etc.) give a negative ORP reading (e.g., −100 to <−1000 mV) [59].

The ORP value of biological fluids is generally reductive, and the ingestion of certain highly oxidizing substances (e.g., peroxides, perchlorates, permanganates, etc.) can be harmful to the body. The proposition that a solution with a negative ORP is healthy and a positive ORP is harmful is reasonable, but faulty [59]. For example, logically, the ingestion of a highly positive ORP solution could be harmful due to the highly oxidizing chemicals; ergo, a solution with a lower ORP is likely to be less harmful. Moreover, if the solution is reductive, as indicated by a negative ORP, then perhaps this would maintain or even restore cellular redox homeostasis. However, the reasoning is faulty because there is no indication that a redox reaction will even occur due to the possibility of insurmountable activation energy, or that the chemical species responsible for the negative ORP is beneficial [59]. Indeed, 2-mercaptoethanol provides a negative ORP, but its ingestion could have fatal consequences. In other words, just because a solution has a reductive ORP does not make it harmless, much less good for health. Unfortunately, the focus subsequently became the redox potential itself as opposed to the chemical species responsible for the negative ORP.

The ORP has also been mathematically transformed to try and remove the influence of pH, which is referred to as the rH_2_ index [89]. This rH_2_ index is based on Nernst’s Law and Gibbs free energy using the Boltzmann constant to make it pH-independent relative to the standard hydrogen electrode (SHE). This is then used to estimate the solution’s antioxidant capacity [1]. Unfortunately, this further emphasized the significance of ORP without considering (i) the chemical specie(s) responsible for the ORP (i.e., healthy, or harmful), (ii) the concentration of those species, or (iii) if the activation energies would even allow a meaningful reaction rate to proceed.

The overt focus on the ORP (e.g., the more negative the healthier) instead of on the concentration of dissolved H_2_ may have been because it was not clear that H_2_ was responsible for producing the negative ORP in ERW, but it was clear that, when the negative ORP was eliminated, the therapeutic benefits were likewise eliminated. Since the redox potential represents the direction of a potential electron transfer, there have been misunderstandings by some that the negative ORP is due to the literal presence of electrons dissolved in the water, perhaps even because the term “reduced” in ERW refers to the electrochemical “reduction” that occurs at the cathode. The implication is that electrons come off the negative platinum electrode during electrolysis and, instead of reacting with protons to form hydrogen gas, they remain dissolved in the water to produce a negative ORP [52,55], perhaps making negatively charged “ionized” water molecules (e.g., H_2_O^−^ or H_2_Oe^−^, H_2_Oe_solv_^−^). However, it would be impossible under normal conditions to have stable solvated/hydrated electrons in the bulk phase of liquid water [61,90,91], and this would be toxic even if it were achieved [92].

Instead, the negative ORP is a direct result of the dissolved hydrogen gas, as shown by the Nernst equation (Equation (1)) according to the standard half-cell reduction potential of hydrogen ions to hydrogen gas (Equation (2))
(1)$$E=\;E{}^{\circ}-\left(\frac{\mathrm{RT}}{\mathrm{nF}}\right)\ln\left(\frac{PH_{2}}{{\left[H^{+}\right]}^{2}}\right)=-\left(\frac{\mathrm{RT}}{\mathrm{nF}}\right)\left(\ln\left(PH_{2}\right)+\left(\frac{2\mathrm{pH}}{\log\left(e\right)}\right)\right)$$
2H^+^ + 2e^−^ ⇆ H_2_ (g)  *E*° ≡ 0.000 V(2)
where *E* is the oxidation–reduction potential, *E*° is the standard half-cell reduction potential (0.000 V), R is the universal gas constant (8.3145 J⋅K^−1^⋅mol^−1^), T is the absolute temperature in Kelvin, F is the Faraday constant, 96485 C⋅mol^−1^, n is the number of moles of electrons transferred in the half-reaction, and *P*H_2_ is the pressure of H_2_ in atm. Therefore, according to Equations (1) and (2), if hydrogen gas is added to water or the pH increases (decrease in H^+^ concentration), the ORP will be negative. Indeed, the removal of hydrogen gas from the ERW results in a positive ORP, and simply bubbling hydrogen gas into water results in a negative ORP [17].

Theoretically, based on the equations, if the pH were held constant or by using the rH_2_ index to make the redox potential independent of pH, the solution with the greater negative ORP would have the highest concentration of H_2_. However, as can be seen in the equations, the influence of pH on the ORP largely obfuscates any influence on the ORP from changes in H_2_ concentration, as illustrated in Figure 1. Figure 1A demonstrates that changes in pH influence the ORP more than changes in H_2_ concentration. Indeed, changing the pH from 0 to 14 changes the ORP by around 800 mV, whereas changing the H_2_ by a factor of ten (0.16 mg/L to 1.6 mg/L) only changes the ORP by 29.58 mV. Similarly, Figure 1B illustrates that even when increasing the H_2_ concentration by 100× (0.14 mg/L to 14 mg/L), the ORP only changes by 59.16 mV, whereas increasing the pH from 7 to 12 (typical of some ERW devices), the ORP changes by 295.8 mV. Importantly, the 59.16 mV ORP change seen with the 100× increase in H_2_ concentration is the same magnitude of change accomplished by only a one-unit change in pH (a 10× change in the H^+^ concentration). In Equation (1), the effect on ORP of a change in pH from 0 to 14 is equivalent to increasing the concentration of H_2_ by a factor of 10^28^ times. In other words, the reason why pH has more of an effect on the ORP than H_2_ in Figure 1 is that pH is the logarithm of [H^+^], the term of which is also squared, whereas the pressure of H_2_ is expressed in a linear scale, ergo it has a smaller dynamic range compared to [H^+^]. For a more detailed discussion on ORP, see [59].

Although the graph gives the impression that even with no H_2_, the ORP would still be negative, this is only because of the wide range of the H_2_ concentration. Importantly, however, based only on the Nernst prediction and without considering any other redox couples, the ORP of water would still be negative (−290 mV) even with a H_2_ concentration of only 0.0001 mg/L [59].

Finally, the negative ORP reading in ERW can be completely explained by the dissolved H_2_ levels and pH, and thus no additional enigmatic explanations such as “free electrons” or “stored energy” are required [59]. As mentioned, many compounds can make a negative ORP solution, some with lethal consequences. The negative ORP in ERW is due to hydrogen gas and not “free electrons”.

### 4.4. Atomic Hydrogen

The idea that atomic hydrogen was responsible for the benefits of ERW started around 1995 [17,27,48]. This idea was proposed because molecular hydrogen was thought to simply be an inert byproduct of electrolysis. However, the existence of atomic hydrogen in ERW is very unlikely because it is a highly reactive free radical and would react with other hydrogen atoms (H^•^ + H^•^→H_2_), whose high rate constant (10^10^ M^−1^ s^−1^) is only limited by the rate of diffusion [61]. Moreover, Hiraoka, et al. specifically investigated this claim and were unable to find any evidence of atomic hydrogen in the water [60]. They concluded that any antioxidant-like effects of the water could easily be explained by usual physico-chemical knowledge without a reliance on “active hydrogen”. There are simply no phenomena in ERW that require the presence of “active hydrogen” [60].

### 4.5. Platinum Nanoparticles and Elemental Mineral Colloids

As more research on ERW continued, it was observed that, in certain cases, ERW acted differently from hydrogen water prepared by bubbling hydrogen gas into water. For example, some, but not all, ERW scavenged the superoxide anion radical and hydrogen peroxide [17,28], which molecular hydrogen cannot do [46]. The electrode degradation and subsequent concentration of platinum nanoparticles (PtNPs) are time-dependent [1,35,79]. It was found that, in some cases where continuous electrolysis was performed for 2 h, ERW contained a small number of PtNPs in the ppb range due to the degradation of the electrodes [22,41,79].

It is known that atomic hydrogen readily adsorbs onto a platinum surface and that PtNPs can catalyze the activation of molecular hydrogen to reactive free radical atomic hydrogen or simply electrons and protons [93]. Thus, a new hypothesis was suggested that PtNPs in ERW activate molecular hydrogen and that adsorbed hydrogen atoms are readily donated to various reactive oxygen species (ROS) [22,33]. It was also suggested that electrolysis reduces mineral cations at the cathode surface and that the water contains stabilized “mineral nanoclusters” with adsorbed atomic hydrogen and “mineral hydrides” [22,78,79,94].

However, there is no credible evidence for stabilized mineral hydrides and elemental mineral nanoclusters [60,95]. Indeed, their existence is unlikely because the reactivity of these chemical species would rapidly reduce aqueous protons to hydrogen gas [e.g., 2e^−^ (aq) + 2H^+^→H_2_ (g)]. Moreover, as seen in Table 2, the standard reduction potentials of a mineral/metal require more than 2 volts [62], which is significantly greater than the reduction of protons to hydrogen gas even at a pH of 14 [96,97]. In other words, protons are preferentially reduced to hydrogen gas at the cathode during electrolysis as opposed to minerals/metals [62].

Similarly, atomic hydrogen formed at the cathode is unable to reduce an alkaline mineral or form a hydride with them while in an aqueous solution based on their standard reduction potentials [61,97,98]. Instead, hydrogen atoms will combine with each other as the termination step in radical reactions. Therefore, the ROS-scavenging ability of ERW containing PtNPs can be explained by the presence of PtNPs alone with no need for adsorbed hydrogen [74,99,100,101].

### 4.6. Superiority of ERW to Hydrogen Water?

Some companies may assume that ERW with the same concentration of hydrogen as a non-ERW hydrogen water has more biological effects (good or bad) due to the presence of PtNPs. For example, one often-cited study sponsored by Nihon-Trim (the largest Japanese water ionizer manufacturer followed by Panasonic) reported that ERW exerts superior reactive-oxygen-species-scavenging activity than the equivalent level of dissolved hydrogen [102]. The study rightfully demonstrated that, due to the presence of PtNPs, the scavenging activity of ERW remained even after the removal of dissolved H_2_ by autoclaving. These PtNPs can either directly scavenge ROS or may even activate the intracellular antioxidant system [102]. Logically, if hydrogen water is supplemented with additional scavengers (e.g., ascorbic acid, PtNPs, etc.), then it is expected to exert a greater scavenging activity than hydrogen alone. Indeed, this study demonstrated that the addition of PtNPs to hydrogen water exhibited greater antioxidant activity than H_2_ water alone [102].

Although there may be some additional benefits to PtNPs in ERW [22], or perhaps a synergistic effect from the presence of the PtNPs [94,103], there are also potentially toxic effects [104], which is one of the safety concerns discussed regarding the risks of ERW ingestion (part II of our tandem reviews). Fortunately, the only waters that appear to have PtNPs are those where either the water was supplemented with PtNPs [44,103] or electrolysis was performed on a NaOH solution continuously for 1–2 h in a batch-type water ionizer [33,35,44,75,76,77,78,79,105]. Indeed, many studies on ERW have shown that PtNPs are not in ERW or are at least below the detection limit (0.1 μg/L) [1,60,106].

Taken together, the addition of PtNPs could reasonably result in a greater biological effect in cells (not necessarily humans). Importantly, this biological effect may be toxic to cancer as well as normal human cells. The toxicity is largely dependent on the size and dose of the PtNPs. Nevertheless, as mentioned, everyday users likely do not have to be unduly concerned about this because of the lack of PtNPs in ERW, since ERW containing PtNPs either has had PtNPs added to it, or the water was subjected to continued electrolysis for 1–2 h [33,35,44,75,76,77,78,79,105].

One other argument that has been made is that ERW contains a nongaseous antioxidant because the therapeutic protective effects persisted even after H_2_ had dissipated out [107]. However, even assuming that this observation was not due to PtNPs, the conclusion is flawed because it incorrectly assumes that H_2_ only exerts direct ROS-scavenging activity. In fact, it is unlikely that H_2_ exerts any significant direct radical-scavenging activity [108,109,110,111]. The biological benefits of H_2_ are best explained by the modulation of signal transduction and gene expression, including the upregulation of endogenous antioxidants [108]. Therefore, H_2_ can have a residual protective biological effect for hours [112], days, and even weeks following its brief exposure [113]. With this understanding, the study in [107] demonstrates once again that H_2_ is the molecule responsible for the therapeutic effects of ERW.

## 5. Homing in on Molecular Hydrogen

As research on ERW continued, it was a logical process to eliminate one variable at a time. In 2002, a research group used a rat model of aspirin-induced gastric mucosal injury to eliminate the non-therapeutic properties of ERW [34]. They made four groups: (i) ERW at a pH of 10.5, (ii) a water with similar pH prepared by NaOH, (iii) a water with similar mineral content as ERW, and (iv) a water with a similar pH and mineral content. Not surprisingly, as per our discussion above, the alkaline pH alone provided no benefits. This was also true for the minerals in the water, which were at a relatively low concentration. However, it was found that only the ERW was effective [34], which further confirmed the importance of dissolved hydrogen gas [114,115].

At least by 2004, magnesium metal media were used by researchers to produce alkaline water, which also exhibited a negative ORP and had anticancer and anti-diabetic effects like ERW [36,116,117]. This was before hydrogen gas was known to be the therapeutic component and so H_2_ was not even mentioned in the cited studies. Elemental magnesium produces hydrogen water and an alkaline pH according to the reaction (Mg + 2H_2_O→H_2_ + Mg(OH)_2_). However, these studies demonstrated that there was nothing unique to the electrolysis process, which narrowed down the variables in determining the therapeutic agent.

### 5.1. H_2_ Is Exclusively Responsible for Any Observed Therapeutic Effects

In 2005, it was demonstrated that neutral-pH hydrogen-rich electrolyzed water could reduce oxidative stress in rats [118], which further confirmed that the molecular hydrogen, not the alkaline pH, was responsible for the therapeutic effects [119]. Other studies on ERW also used neutral-pH water [42,50,89,120,121,122,123]. Similarly, in 2006, some ERW researchers began referencing in their studies that H_2_ was the physiologically active substance [42,106,124], as opposed to trace reductive metals [106]. Several years later, researchers used the magnesium metal method to make alkaline water with a negative ORP but removed most of the H_2_ from the water. They demonstrated that the therapeutic effect was the dissolved hydrogen and not the magnesium, ORP, or alkaline properties [125,126].

In 2013, researchers tested filtered water, ERW, and degassed ERW. As expected, the therapeutic effects were only observed using the non-degassed ERW [127]. A more detailed study examined different concentrations of H_2_ at varying pHs in a rat model of aspirin-induced gastric injury [114]. They created four different types of water: ERW at pH 8.5 and 9.5, both of which had either additional H_2_ gas bubbled in to increase the concentration or nitrogen gas to remove the H_2_ gas (as a control). It was found that the lowest H_2_ concentration (0.07 mg/L) was not effective, but the higher H_2_ concentrations (0.22 mg/L and 0.84 mg/L) were dose-dependently effective. Furthermore, when the H_2_ concentrations were the same, a pH of 8.5 was equally as effective as a pH of 9.5 [114]. Similarly, ERW produced by a conventional water ionizer (H_2_—1.3 mg/L, ORP −772 mV, pH—10.8) was used to treat cancer in vitro and in vivo. The control water was ERW with the H_2_ gas removed. For the in vitro treatment, the pH of the ERW was neutralized. The H_2_ in the ERW inhibited cancer cell survival and induced apoptosis. For the in vivo study, it was found that, compared to the non-H_2_ ERW, treatment with the ERW containing H_2_ significantly delayed the development of mammary tumors in transgenic BALB-*neu*T mice [128].

Finally, in 2019 we published a study that further clarifies and confirms that molecular hydrogen is the sole agent responsible for the biological effects of ERW in an animal model of non-alcoholic fatty liver disease (NAFLD) induced by a high-fat diet [2]. Briefly, in this study, we used ERW which had a pH of 11, an ORP of -495 mV, and an H_2_ concentration of 0.2 mg/L. We then divided the mice into four groups: (i) regular diet (RD)/regular water (RW); (ii) high-fat diet (HFD)/RW; (iii) RD/ERW; and (iv) HFD/ERW. The weight and body composition of the mice were measured. After twelve weeks, the animals were sacrificed, and their livers were processed for histology and reverse-transcription polymerase chain reaction. At the time, our results were interesting because, in contrast to other studies, our results showed no differences between the groups drinking ERW and RW in either the RD or HFD. We hypothesized that the null result was due to a low H_2_ concentration. Therefore, we evaluated the effects of RW and low and high HRW concentrations (L-HRW = 0.3 mg/L and H-HRW = 0.8 mg/L, respectively) in mice fed an HFD. Compared to RW and L-HRW, H-HRW resulted in a lower increase in fat mass (46% vs. 61%), an increase in lean body mass (42% vs. 28%), and a decrease in hepatic lipid accumulation (*p* < 0.01). We also found that exposure of hepatocytes isolated from mice drinking H-HRW to palmitate overload demonstrated a protective effect from H_2_ by reducing hepatocyte lipid accumulation in comparison to mice drinking regular water. Accordingly, we concluded that indeed H_2_ is the therapeutic agent in electrolyzed–alkaline water responsible for the effects of attenuating HFD-induced nonalcoholic fatty liver disease in mice [2].

After the 2007 *Nature Medicine* article by Ohsawa et al. demonstrated the overt benefits of molecular hydrogen [46], many ERW researchers began focusing exclusively on hydrogen, and the use of water ionizers became a means of producing hydrogen-rich water [80,122,123,127,129,130,131,132] as opposed to a means of producing ERW. It was noticed that virtually none of the pre-2007 articles reported the hydrogen concentration (or even mentioned hydrogen), but nearly all the post-2007 articles did. Table 3 summarizes some of the studies that helped conclusively demonstrate that molecular hydrogen is the exclusive agent responsible for both the negative ORP and the observed biological effects.

In 2021, Nihon-Trim sponsored another study demonstrating that neutral-pH hydrogen water had the same hepatocellular protective effect against ethanol toxicity as ERW [133]. They also reported that the cytoprotective and ROS-reducing effects of ERW were abolished by degassing but not by neutralizing the pH, confirming that H_2_, not the pH or alkalinity, is responsible for the therapeutic effects of ERW [133].

As can be seen, many studies have demonstrated that molecular hydrogen is responsible for the benefits of ERW and not the other properties. This observation is illustrated in Figure 2, which is a graphical representation of what we would expect when collectively studying the individual characteristics of ERW. To summarize, the control (filtered water) indicates the baseline level. This level is similar to ERW at pH 10, but with the H_2_ degassed from the water. If anything, the level is slightly below baseline, indicating a potentially harmful effect when pH exceeds 10. Additionally, this illustrates that there are no “magical” effects due to electrolysis that were somehow imparted into the water, including microclustering, electrons, energy, etc. Next, as expected, providing an alkaline pH 10 water produced with a hydroxide salt (e.g., sodium, potassium, calcium, etc.) is not different from control except for maybe, again, a slightly harmful effect attributed to the very high pH. Adding a small amount of minerals (<50 mg/L) such as calcium and magnesium to the water also does not provide any obvious benefit. We next notice a non-significant beneficial trend of pH 10 ERW when there is a trace amount of H_2_ (≈0.05 mg/L). As mentioned, this is enough H_2_ to provide a very negative ORP but is not enough to provide any therapeutic effect. However, when ERW is provided that contains a higher concentration of H_2_ (i.e., 1 mg/L), therapeutic effects are observed. This same benefit is observed when neutral-pH water has H_2_ gas bubbled into it to the same concentration. There might be a minor trend of a greater benefit from the neutral pH, perhaps due to the potential harmful effects of such a high pH from the ERW (see part II of our tandem reviews).

Finally, the last example column in Figure 2 illustrates a frequently observed (although not always), dose-dependent effect of H_2_ concentration and a corresponding therapeutic effect. In this hypothetical example, the neutral-pH water containing more H_2_ than either the ERW at pH 10 or neutral-pH water had a greater protective effect. In summary, this simulated figure further illustrates (i) that H_2_ is exclusively responsible for the benefits in ERW, (ii) that the H_2_ concentration must be high enough to exert a beneficial effect, and (iii) that there can be a dose-dependent therapeutic effect of H_2_.

### 5.2. Broad Recognition of the Importance of H_2_ Still Needed

Despite the overwhelming research that conclusively demonstrates that H_2_ is exclusively responsible for the therapeutic effect of ERW, it is still sometimes overlooked by researchers and the industry. This is illustrated by searching for ERW articles on PubMed that either contain or do not contain hydrogen in the title or abstract (See Figure 3). Prior to 1988, there were no ERW articles that mentioned hydrogen. This brief analysis shows that, although the ratio of articles that do not mention hydrogen to the total number of ERW articles is gradually decreasing, more than 50% of ERW articles still do not address hydrogen. For example, several review articles and clinical studies on ERW discuss the importance of molecular hydrogen but do not explicitly recognize that the benefits are due exclusively to molecular hydrogen [134,135,136,137]. Furthermore, other review articles still do not recognize the importance of H_2_ [138] or even incorrectly assume the benefits are due to the alkaline pH [139].

This unnecessary confusion may result in studies using ERW where the H_2_ concentration is not even measured. For example, a randomized controlled study found that 8-week consumption of 2 L/day of alkaline ionized water decreased markers of oxidative stress, and improved quality of life and exercise performance [140]. However, the concentration of molecular hydrogen was not reported, only the ORP, which cannot be used to estimate the concentration of dissolved H_2_ [59]. This makes it impossible to know if the results were based on the daily ingestion of high H_2_ or low H_2_, since we do not know the initial concentration, and the water was only prepared twice a week [140]. Similarly, another study tested the effects of filtered water, pH 8.0, pH 9.5, and a combination of pH 9.5 and 11.5 on the levels of blood sugar in diabetic patients for 14 days [52]. Despite being alkaline with a negative ORP, the pH 8 water had no beneficial effects compared to the filtered control water. However, pH 9.5 and 11.5 had dose-dependent therapeutic effects. The results are consistent with our understanding of H_2_ in ERW. The pH 8 alkaline water likely did not provide a sufficient concentration of H_2_, whereas the pH 11.5 water had more H_2_ than the pH 9.5 alkaline water, explaining the dose-dependent effects. Unfortunately, the study never measured the concentration of molecular hydrogen and only referenced other studies on hydrogen water, and the benefits of ERW were attributed to the debunked ideas of atomic hydrogen, microclustering, alkalinity, and electrons [52]. Therefore, it is hoped that the essential importance of molecular hydrogen in ERW will be more widely recognized by researchers and consumers. This will allow for better research designs, more meaningful findings, and better interpretations of the results. Additionally, although the true clinical effects of hydrogen water remain inconclusive, it is important that potential customers at least know the facts. This way they can be guided as to which product may be the best for them, without getting confused by these many debunked claims surrounding ERW.

Additionally, as detailed in part II of our tandem reviews, ERW has some potential health concerns when ingesting alkaline water, specifically ERW at a pH above 10. For example, (i) increased degradation of the electrodes and subsequent release of PtNPs and/or other potentially toxic metals causing tissue damage, (ii) impaired mineral and nutrient absorption, (iii) dysregulated microbiome, (iv) impaired potassium homeostasis, etc. This is problematic since the highest concentration of H_2_ in ERW is only obtained with the concomitant production of a very high pH. Moreover, as is also discussed in part II, due to differences in ERW machines, source water, and flow rate, the production of H_2_, even at the highest setting, may be less than what is used in successful clinical studies. Although these concerns and caveats can be circumvented, many are not even aware of the importance of H_2_, let alone that these aspects are problematic. Indeed, the idea that high-pH ERW and PtNPs are what is therapeutic is akin to the belief that the antidote is the poison. Molecular hydrogen needs to be measured in ERW, and, as discussed and extensively reviewed [59], ORP meters cannot be used to estimate the concentration of H_2_ in the water. Unfortunately, the problems with ORP meters extend to commonly used and marketed portable H_2_ meters. These types of H_2_ meters are not calibrated but are based on measuring the ORP and then using an algorithm in an attempt to calculate the H_2_ concentration [59]. However, these ORP-based H_2_ meters have been shown to be inaccurate, especially at higher and lower pH values, in which the H_2_ concentration is significantly over or underestimated, respectively [59].

## 6. Conclusions

ERW has been used for many decades, but it was not until recently that it was conclusively demonstrated that molecular hydrogen is responsible for both the negative ORP and the observed biological benefits. A negative ORP may indicate the presence of H_2_, but due to the dominating influence of pH, the magnitude of a negative ORP reading cannot be used to estimate or compare the concentrations of hydrogen water. Importantly, the presence of H_2_ alone can fully explain the benefits of ERW without requiring any need to rely on inaccurate or metaphysical concepts such as alkalizing the body with alkaline water, microclusters, free electrons, atomic hydrogen, increased entropic energy, etc. Although the safety of H_2_ is well established, there remain important safety concerns with ERW (see part II of our tandem reviews). However, these safety concerns may go unrecognized when they are seen as the antidote due to a lack of understanding about the role of molecular hydrogen in ERW. Future research on ERW should be sure to measure the concentration of H_2_ using an accurate method and recognize the importance of H_2_ when designing a study and interpreting the results. Consumers should be leery of companies promoting ERW with claims that either go beyond or contradict the scientific data.

## Figures and Tables

**Figure 1 ijms-23-14750-f001:**
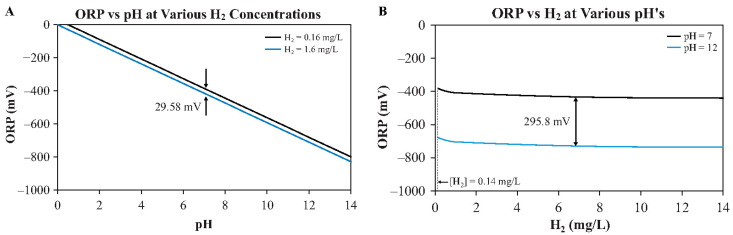
Changes in ORP as a function of either pH (**A**) or H_2_ concentration (**B**). Temperature held constant at 25 °C. Note that although the concentration of H_2_ at the standard ambient temperature and pressure is 1.6 mg/L, it may be higher than this at lower temperatures and/or higher pressures accordingly to Henry’s law (C = P/K_H_; C is concentration, P is pressure, and K_H_ is Henry’s solubility constant for the specific gas at a given temperature).

**Figure 2 ijms-23-14750-f002:**
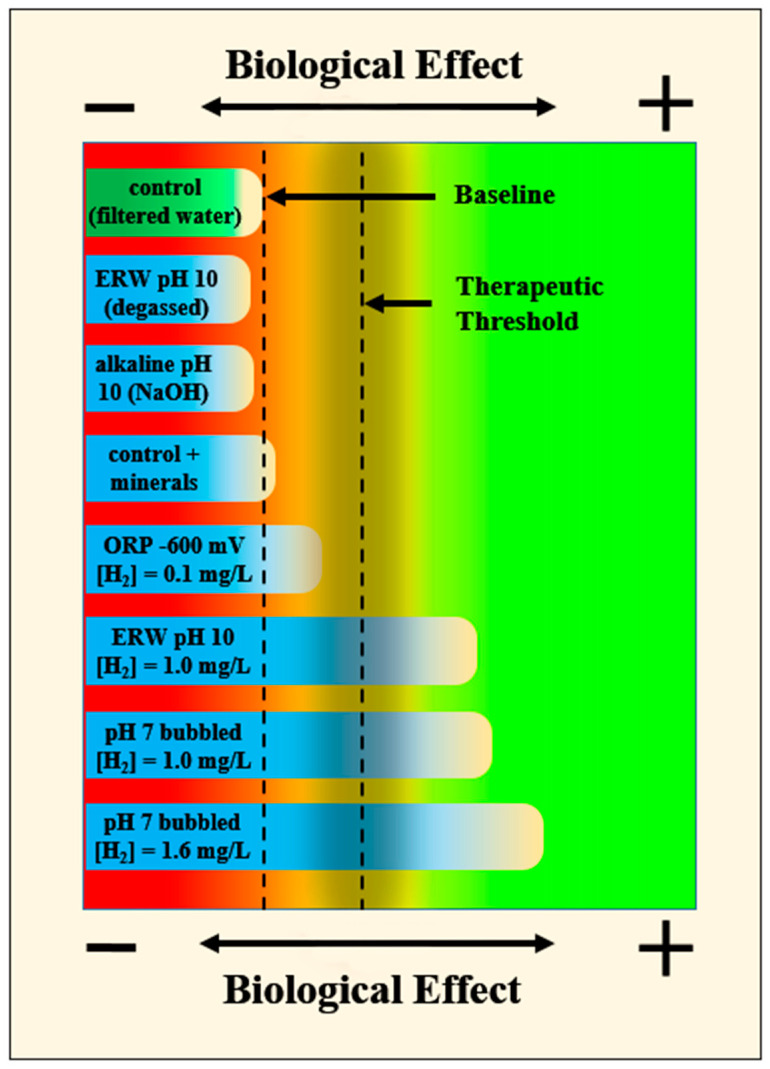
A hypothetical symbolic representation of relative benefits from various water conditions. ERW or NaOH pH 10 water provides no benefit compared to baseline with a downward trend. A negative ORP from the presence of a low level of H_2_ exerts only a positive beneficial trend. However, at higher H_2_ concentrations in ERW or with bubbled H_2_, the therapeutic threshold is passed, and there is a dose-dependent H_2_ benefit.

**Figure 3 ijms-23-14750-f003:**
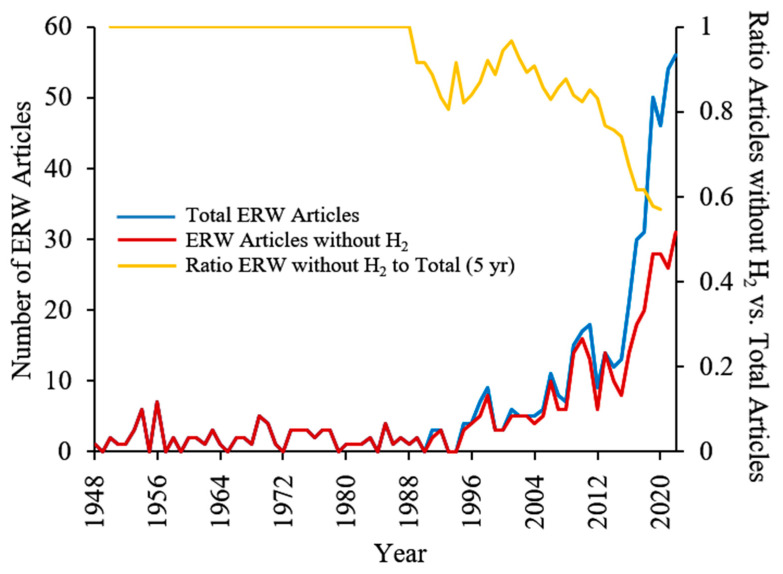
The annual number of ERW articles with or without hydrogen in the title or abstract from 1948 to 2022. ERW articles are defined as having “alkaline (ionized) water” or “electrolyzed reduced water” in the title. The presence or absence of “hydrogen” was searched for either in the title or abstract. The ratio of ERW articles without hydrogen to the total number of ERW articles represents an average of 5 years.

**Table 1 ijms-23-14750-t001:** Frequently claimed physico-chemical properties of ERW.

Claimed Physio-Chemical Properties of ERW	References	Comment	References
Falsifiable			
Contains high levels of dissolved hydrogen gas	[22]	H_2_ level varies significantly, but it is the exclusive reason for the benefits	[2]
Contains high levels of dissolved oxygen gas	[57]	No, contains less O_2_, but even if it had more, it would not be therapeutic	[22]
Contains platinum nanoparticles (PtNPs)	[22]	Unlikely except for (1) continuous electrolysis and (2) when high voltage is applied. PtNPs may be toxic	[58]
Negative oxidation–reduction potential	[22]	Yes, from the dissolved H_2_ gas	[59]
Contains “active” atomic hydrogen	[22]	No, scientifically impossible and has been directly investigated and refuted	[60,61]
Contains mineral hydrides	[22]	No, scientifically impossible and has been directly investigated and refuted	[60,61]
Contains abundant free electrons	[52,55,57]	No, scientifically impossible and has been directly investigated and refuted	[60,61]
The “hydroxyl ions” are the cloudy antioxidants in ERW that produce the negative ORP	[57]	No, OH^−^ (hydroxide) ions make the pH alkaline, the cloudiness is H_2_ gas, and OH^−^ is not an antioxidant; indeed, removing an e^−^ would make it the most reactive hydroxyl radical (^•^OH)	[62]
Minerals in ERW are more bioavailable	[57]	No, in fact, they may be less bioavailable because the alkaline pH reduces their solubility, which is why there are often calcium precipitates in ERW.	[62]
ERW boils and freezes significantly differently from normal water	[57]	No, if the solute and ion concentration are the same per molal boiling point elevation and freezing point depression. Has been investigated and refuted.	[62]
Altered water structure, different hydrogen bond angle, hexagonal water, microclustered water, etc.	[52,57,63]	No, scientifically impossible in the context of bulk liquid water. The claim has also been directly investigated and refuted	[47,51,64,65]
Reduced surface tension	[57]	No, additionally pH does not significantly influence surface tension. This claim has also been directly investigated and refuted	[66]
Electrically charged as indicated by negative ORP	[57]	No, ERW is electrically neutral (obeys the law of electroneutrality); the negative ORP is due to H_2_ gas (see text)	[62]
The alkaline pH is responsible for the benefits	[55,57]	No, scientifically implausible and many favorable ERW studies have specifically refuted this claim	[2]
Unfalsifiable			
ERW is energetically enhanced	[57]	Unknown what these types of claims mean. However, since the benefits of ERW are eliminated once H_2_ is removed, then the exact meaning or number of these metaphysical claims is irrelevant.	
ERW is “imbued with frequencies” during the process of electrolysis	[67]		

**Table 2 ijms-23-14750-t002:** Redox potentials for various minerals/metals.

Half-Cell Reduction Reaction			E° (V)
Pt^2+^ (aq)	+2e^−^	→Pt (s)	1.18
Pd^2+^ (aq)	+2e^−^	→Pd (s)	0.92
Ag^+^ (aq)	+e^−^	→Ag (s)	0.80
Fe3^+^ (aq)	+e^−^	→Fe^2+^ (aq)	0.77
Cu^2+^ (aq)	+2e^−^	→Cu (s)	0.34
* 2H^+^ (aq)	+2e^−^	→H_2_ (g)	0.000
^†^ 2H^+^ (aq)	+2e^−^	→H_2_ (g)	−0.83
Na^+^ (aq)	+½H_2_ (g) +e^−^	→NaH (s)	−2.37
Mg^2+^ (aq)	+2e^−^	→Mg (s)	−2.38
Na^+^ (aq)	+e^−^	→Na (s)	−2.71
Ca^2+^ (aq)	+2e^−^	→Ca (s)	−2.87
St^2+^ (aq)	+2e^−^	→St (s)	−2.89
Li^+^ (aq)	+e^−^	→Li (s)	−3.04

***** Standard ambient temperature and pressure (pH = 0). ^†^ pH= 14, temperature = 298.15 K, pressure = 1 atm.

**Table 3 ijms-23-14750-t003:** Summarized evidence demonstrating H_2_ is the therapeutic agent in ERW.

Procedure	Conclusion	References
Aspirin-induced gastric mucosal injury. Groups: (1) ERW, (2) same pH, (3) same minerals, (4) same pH and minerals	Only ERW was effective, demonstrating that molecular hydrogen, not the minerals or pH, is what is important in ERW	[34,114,115]
Neutral-pH ERW was provided to rats injected with a free radical inducer, 2-azobis-amidinopropane dihydrochloride	Despite the neutral pH, ERW exerted significant antioxidant protection, which indicates the importance of molecular hydrogen	[118]
Studies neutralized pH before adding to cell culture	Eliminates the alkaline pH property from contributing to the benefits	[29,33,35,40,43,44,73,74,75,76,77,78,79,80,81,82,89]
Animal studies in which neutralized pH ERW was given with high levels of H_2_.	Eliminates the alkaline pH as a contributor to the benefits	[35,92,93,94,133]
Water with a negative ORP produced with magnesium metal instead of electrolysis.	Eliminates any “magical” properties induced by electrolysis, while ensuring the presence of H_2_	[125,126]
High-fat diet-induced liver disease. Groups: (1) control, (2) low-H_2_ ERW, (3) high-H_2_ ERW	Only the high H_2_ ERW group had any benefits, despite the low H_2_ also having an alkaline pH and negative ORP	[125,126]
Use of ERW with high pH, but an ORP that was barely negative (e.g., only −200 mV)	No observed benefits because the level of H_2_ was neither high enough to give a more negative ORP nor to provide any biological effects.	[2,52] multa nimis *

* multa nimis (far too many) published and unpublished observations by investigators and consumers on the use of ERW.

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
