# Peer review of "Electrolyzed–Reduced Water: Review I. Molecular Hydrogen Is the Exclusive Agent Responsible for the Therapeutic Effects"

_ijms, 2022, doi:10.3390/ijms232314750_

Round 1

Reviewer 1 Report

This article is well written, organized correctly and examines electrolyzed reduced water (ERW) and all the myths and benefits from a scientific perspective. Introduction and Hydrogen in Alkaline Ionized Water are reasonable given the premise of the paper. The structure in Rejected hypotheses on ERW, Homing in on Molecular Hydrogen and Conclusion parts are well argued, the logic easy to follow and the benefits of the analysis of the previous papers are obvious and successful. Figures and tables are very significantly helpful and comprehensible throughout the text.

The paper aims an extend analysis of previous papers about electrolyzed reduced water and molecular hydrogen, busting myths and changing the way things are considered. The critical way of analysis is well argued and very thorough also.   

Although, there are some really minor comments, that they want some attention:

  1. The title is not very given for the content, it is more the result of the whole study/review. Also, given that the article is mainly a review, it is maybe a good practice to be depicted that point in the title.
  2. On page 8 in lines 312-313 of the article, there is a mistake with bracketing and the equitation, I believe. There is no close of the bracketing and the eq is 1 not 3 (or you must change the numbers).
  3. On page 3 of the article, the part 3 of the paper Rejected hypotheses on ERW has a false format according to equivalent other parts (Introduction, Hydrogen in Alkaline Ionized Water, etc.).

Author Response

Thank you for your thoughtful comments and approval of our article. We appreciate the suggested edits, which we respond to below.

Comment 1-1

The title is not very given for the content, it is more the result of the whole study/review. Also, given that the article is mainly a review, it is maybe a good practice to be depicted that point in the title.

Answer 1-1

Thank you for your suggestion. We added “Review” in the title to explicitly indicate that this is a review article.

Comment 1-2

On page 8 in lines 312-313 of the article, there is a mistake with bracketing and the equitation, I believe. There is no close of the bracketing and the eq is 1 not 3 (or you must change the numbers).

Answer 1-2

Thank you for noticing the error. Initially, we tried to show some of the steps of modifying the Nernst equation, but for simplicity we omitted the middle step. We also omitted the initial general form of the Nernst equation since we do not discuss this in the text, but we do provide a reference that discusses the details of this in depth (ref 48). Whereas before Eq 1 was written as:

We have shortened it as follows:

Comment 1-3

On page 3 of the article, the part 3 of the paper Rejected hypotheses on ERW has a false format according to equivalent other parts (Introduction, Hydrogen in Alkaline Ionized Water, etc.).

Answer 1-3

Thank you for noticing this formatting inconsistency. We changed the format as suggested.

Reviewer 2 Report

Dear Editor,

The manuscript "Electrolyzed Reduced Water: I. Molecular hydrogen is the Exclusive Agent Responsible for the Therapeutic Effects" describes the last works performed on electrolyzed reduced water (ERW). The manuscript discusses many topics related to the properties of ERW and the common misunderstanding of some parameters related to its quality. The manuscript is well-structured and written in the sound language. However, I included some suggestions to improve the manuscript in the pdf file of the manuscript in the form of notes.

Although the authors treated well the different hypotheses about the effectiveness of ERW and concluded that the biological benefits of ERW are related only to the presence of H2, it will be useful for readers to have some information about the properties of H2. I suggest adding a section about the physicao-chemical and biological properties of H2. This will help readers to understand better the discussion in the paper. For example, the lipophilicity of hydrogen allows it to better pass the cell membranes and organelles. Additionally, the claim that the physiological benefits of H2 are only related to the modulation of signal transduction and gene expression is not accurate. In food studies where there is no effect of modulation of signal transduction and gene expression on the quality of foods, we found that H2 could protect the antioxidant activity, phytochemicals, color, and overall quality of different foods during drying and packaging under the H2 atmosphere. The obtained results are surely related to the physicochemical and antioxidant properties of H2.

There are many other suggestions reported inside the pdf file where the authors can reach them.

Author Response

Thank you for your thoughtful comments and suggestions. We have made the following changes per your suggestions, and/or provided our responses below.

Comment 2-1

Italicize “in vitro” and “in vivo”.

Answer 2-1

Thank you for the suggestion. We have italicized all uses of in vivo and in vitro.

Comment 2-2

As you treat the concept of “alkaline water”, I suggest to use the equations related to alkaline water electrolysis (AEM) instead of the used equations of PEM because in the former method hydroxide ions responsible for the alkaline property of water are formed. the following equation is more appropriate:

2OH-   ---->  H2O + 1/2O2 + 2e-

Answer 2-2

Thank you for your suggestions. First, we only wanted to define and introduce water electrolysis in general (using the normal electrolysis equations) and then introduce how the alkaline water is made. We agree about the more appropriate equation for the anodic reaction as it also matches our written text better of oxidation of hydroxides. However, we maintained our original stoichiometry.

Comment 2-3

I suggest to mention briefly some reasons (factors) behind the decrease of pH of body (extrinsic and intrinsic).

Answer 2-3

Thank you for your suggestion. We indicated how CO2 is produced in normal metabolism. We also indicated examples of diseases when pH is critically decreased.

Comment 2-4

Your comment on “bad”.

Answer 2-4

Regarding your suggestion with “bad” and “alkalizing agent” it is apparent that the meaning of our original statement was not clear. We revised the sentence. It now reads: “However, instead of recognizing that alkaline water is an ineffective “alkalizing” agent, they shift the focus to how “acidic” (and therefore “bad”) the beverage is for the body” (Lines 245-247).

Comment 2-5

“The reality is that the organic molecules in tea, like cabbage and many other foods, serve as natural pH indicators.” This explanation and pH-dependent phenomenon is not the only reason. In several studies that we tested the effect of hydrogen-rich water, hydrogen-rich ethanol, hydrogen-rich methanol and hydrogen-rich hexan on the extraction of phytochemical of several vegetables, fruits and oils, we found for all solvents and all plants and oils a significant increase in concentration of these phytochemicals compared with non hydrogen solvents. How can you explain this phenomenon? The change of pH was little between hydrogen-rich solvents and non hydrogen ones.

Answer 2-5

Thank you for your comment. Our statement was primarily focused on the most visual phenomena of the appearance of making tea with alkaline water. This color change in tea occurs with alkaline water that contains no detectable levels of hydrogen gas. Indeed, using sodium hydroxide has the same effect. Bubbling H2 gas into neutral-pH ddH2O water, and then doing the same “experiment” of placing tea bags into this water, gives no “visual” indication of any change of color or appearance of tea. Our statement does not deny the possibility that aqueous H2 increases the extraction ability of phytochemicals. It only indicates that we are not required to believe the visual change is due to so-called microclustering.  Moreover, it is beyond the scope of this paper to discuss the eccentricities of the effects of H2 on solute extraction. It is also possible that this effect is not exclusively due to unique properties of H2, but is common with other dissolved gases as noted in the literature with O2, N2, etc. However, we modified our statement to include that dissolved gases may also contribute to this effect, but we want to be careful since this demonstration is often done with water that contains no detectable levels of dissolved H2. However, to avoid confusion, we have added a sentence stating that gases themselves may also contribute to phytochemical extraction (Lines 295-296).

Comment 2-6

Which effect? There is no information in the previous sentence.”

Answer 2-6

Thank you for your comment. We clarified our sentence to make it clear that we were indeed referring to the previous sentence about the ability of the alkaline water to “mix” with oil.

Comment 2-7

Write it as +500 and >+1000 mV

Answer 2-7

Thank you for your suggestion. We added the + sign in front of 500 and 1000 mV (Line 329)

Comment 2-8

It is important to know this statement is not always true because sometimes we add weak or moderate reducing agents and the ORP value stays at positive range. This is the case of many antioxidants.

Answer 2-8

Thank you for your comment. We agree that many (perhaps most) antioxidants will not give a negative ORP when added to water. We modified our statement by adding the qualifier “certain” before antioxidants (line 330). The other reductive species we listed in parenthesis (i.e., H2, ascorbate, 2-mercaptoethanol) do all give a negative ORP.

Comment 2-9

The aim of  ingestion of reducing compounds such as vitamins, antioxıdants and HRW is not to break covalent bonds of molecules and form new compounds. The importance of ingestion of these reducing agents is to keep the redox homeostasis of the cytoplasm and different fluids in the body as well as neutralization some harmful free radicals.

Answer 2-9

Yes, we agree that the purpose and benefits of reductive compounds is not to directly break covalent bonds, but rather improve the redox status of the cell (e.g., stimulating the production of our body’s natural antioxidants). Our statement does not contradict this truth, but rather illustrates that just because something has a negative ORP does not mean that it will exert any biological activity at all. For example, even water contains electrons that can and do participate in radical reactions, but just because it has electrons, we don’t consider water to be an antioxidant or a free radical. By the same token, just because water has a negative ORP does not mean it will have an antioxidant effect in the body. There are many things that must be considered, which we listed several of them including the activation energy, Gibbs free energy, etc., and finally if the substance making the negative ORP is even beneficial. Another example is the superoxide radical, which although is often called an “oxidant”, it is actually a powerful reducing agent and toxic at high levels. Finally, although our statement was not about antioxidants/vitamins and we agree that many “antioxidants” improve the redox status of the cell without directly breaking covalent bonds, it must also be pointed out that in the end, the redox status is only improved (changed) once there is a literal transfer of electrons, which means that some bonds were broken and others were made (e.g., GSSG to GSH). Something was oxidized and something else was reduced, a literal redox chemical reaction must have occurred. We appreciate your comment, and we hope that this discussion clarifies our original statement.

Comment 2-10

No equation 3

Answer 2-10

Thank you for noticing our error with referring to equation 3, which does not exist. We further modified and simplified this section as seen in lines 380.

Comment 2-11

Replace by “water”. I think it is better to compare water with water not water with gas. So, I suggest to compare ERW with HRW (H2 bubbled water).”

Answer 2-11

Thank you for your suggestion. You are correct that we are specifically referring to when the H2 gas is bubbled into the water to make aqueous H2-water. We revised this sentence as shown in lines 435.

Comment 2-12

The unit of ORP (Eh) is volte or millivolte!

Answer 2-12

We removed “electron” to avoid confusion. Although the relationship is the same since this is dealing with standard conditions (molar concentrations) and free energy change with single electron transfer as the stoichiometry.

Comment 2-13

This statement contradict many research that proved the anti-radical activity of H2, especially the famous study of Ohsawa et al (2007). It is better to use references of other researchers not yours. When we defend an idea against other ones we cannot use our articles as an argument.

Answer 2-13

We understand the concern; however, our statement does not disagree with the Ohsawa et al paper. His paper does not claim that H2 provides its benefits by significantly directly scavenging radicals. The paper shows that under the right conditions H2 can scavenge OH radicals, and according to the authors (personal communication), they were very careful to not claim that the neuroprotective benefits were from H2 scavenging OH radicals. We cited our article because it nicely summarizes the main scientific arguments (and other studies) about the biological insignificance of any potential radical scavenging effect of H2. This is largely based on the rate constants of 2nd order-reaction kinetics juxtaposed with the more abundant nucleophilic targets for the hydroxyl radical. However, to help the reader, and per your suggestion, we also cited other papers (Penders 2014, Li 2021, and even Ohta 2011) that discuss this issue.

Comment 2-14

You did not explain why ERW pH 10, H2 = 1.0 mg/L is lower biologically active than water pH 7, bubbled H2=1.0 mg/L. You need to provide explanation or reference for this claim. In the text, you say the neutral pH water contains higher H2 than the neutral pH water, and in the Figure 2 these two water contain the same H2 level!!!! How is it?

Answer 2-14

We addressed this in the section before the Fig 2 image lines 611-612, which states: “There might be a minor trend of greater benefit from the neutral pH, perhaps due to the potential harmful effect of such a high pH from the ERW.” For additional clarity, we added a comment to refer the reader to part II of our tandem review.

Comment 2-15

I suggest to add a section about the physical, chemical and biological properties of H2. This will help readers to understand better the discussion in the paper. For example, the lipophilicity of hydrogen allowing it to better pass the cell membranes and organelles.

Answer 2-15

Thank you for your suggestion. We have included a section that discusses the chemical properties and biological effects of molecular hydrogen, which is now section 2.0.
